**Brief Communication**

# Europeans support large carnivore recovery while opposing both further population growth and hunting

Guillaume Chapron [1] ✉, Yaffa Epstein [2,3,4], Jeremy T. Bruskotter[5] &
José Vicente López-Bao [6]

After centuries of persecution in Europe, large carnivores are now recovering. Whether this conservation success continues depends in part on public support. Here we show, using a survey of 10,000 respondents across European Union Member States, that while support for the recovery of wolves, bears and lynx remains strong, most respondents oppose both further population growth and hunting—particularly of wolves. Attitudes are remarkably consistent across rural and urban populations, and many respondents express no strong position, suggesting that large carnivores are less polarizing than often portrayed. The recent amendment to the Habitats Directive, which grants Member States greater flexibility to manage their wolf populations, appears broadly aligned with public opinion, as long as its implementation does not lead to population declines. However, the presence of views in tension—rejecting both population growth and hunting— may pose challenges for designing policies that are both science based and supported by the public.

After the cessation of centuries-long eradication efforts throughout Europe, large carnivores have made a remarkable return over the past few decades[1]. Wolves (*Canis lupus*) now number an estimated 19,000 individuals across the European Union (EU)[2]. Other large carnivores are also recovering: the European continent hosts approximately 9,000 Eurasian lynx (*Lynx lynx*) and 20,500 brown bears (*Ursus arctos*)[3]. These recoveries were made possible in large part by strong legal protections that limited human-caused mortality[4]. Large carnivores are protected under the Bern Convention and the EU Habitats Directive, which implements the former in EU law. The Directive requires all EU Member States to ensure that species listed in its annexes achieve and maintain 'favourable conservation status'—meaning populations are viable over the long term, the species' ranges are stable or expanding and sufficient habitat continues to exist to support them[5]. However the strict legal prohibition against killing wolves

that facilitated their recovery in many parts of Europe has now been rolled back: in June 2025, the wolf was moved from Annex IV (strict protection) to Annex V (permitting regulated exploitation) across all Europe (the legal status did not change in the few countries where the wolf was already in Annex V). Our survey—made prior to down-listing the wolf—of 10,807 EU residents from both rural and urban areas across all Member States hosting large carnivore populations (Supplementary Table 1) finds that there remains broad support for the recovery of large carnivores but that letting their populations grow and hunting them are both opposed.

In every country surveyed, we find support for large carnivore recovery outweighs opposition—often by a substantial margin. The strongest support is found in Southern and Eastern Europe (Fig. 1 and Extended Data Fig. 1) and the presence of abundant large carnivore populations does not appear to diminish public support. For example,

[1]Department of Ecology, Grimsö Wildlife Research Station, Swedish University of Agricultural Sciences, Riddarhyttan, Sweden. [2]Department of Law, Uppsala University, Uppsala, Sweden. [3]Swedish Collegium for Advanced Study, Uppsala, Sweden. [4]Center for Advanced Study in the Behavioral Sciences, Stanford University, Stanford, CA, USA. [5]School of Environment and Natural Resources, The Ohio State University, Columbus, OH, USA. [6]Biodiversity Research Institute, CSIC–University of Oviedo, Mieres, Spain. ✉e-mail: guillaume.chapron@slu.se

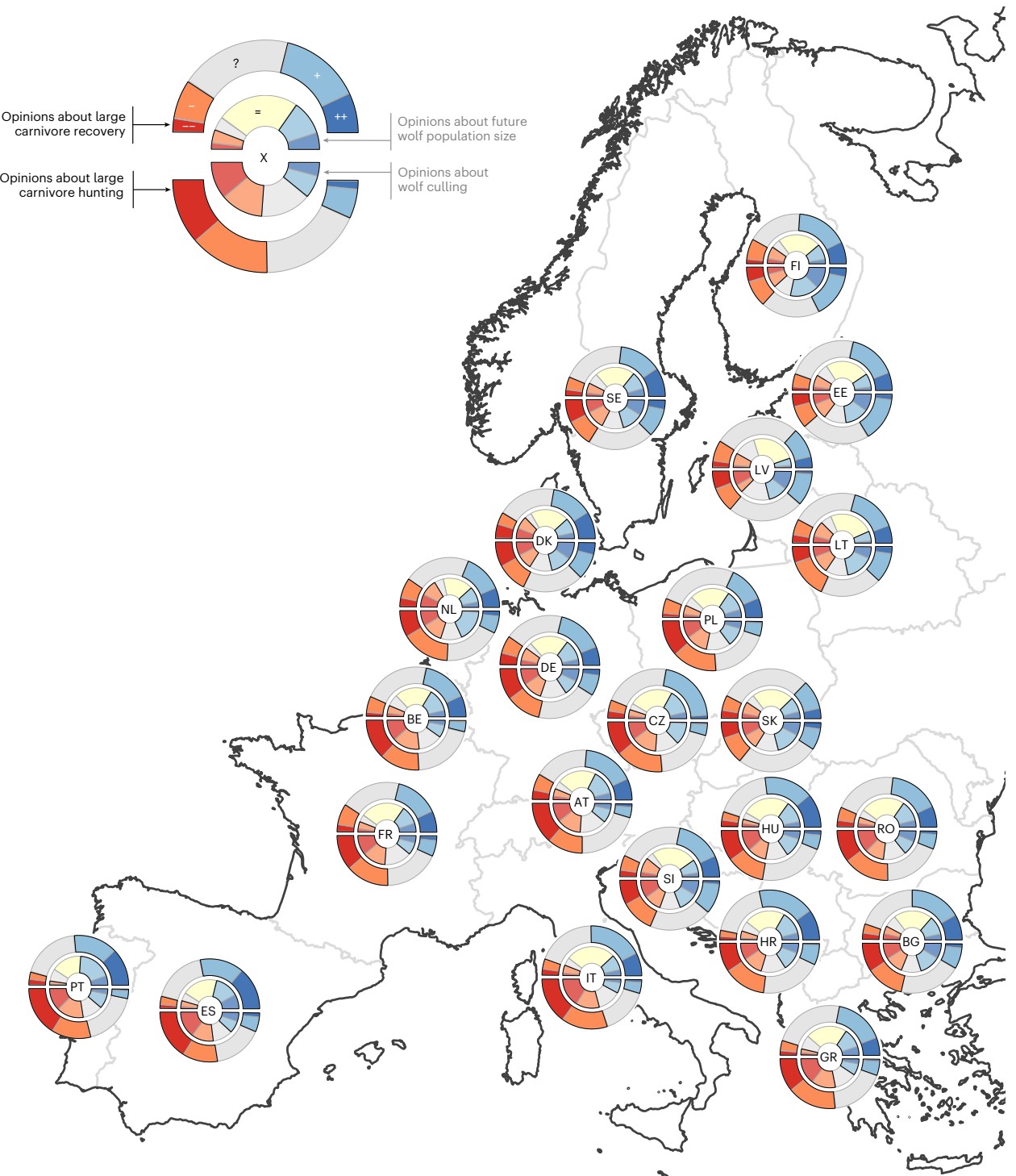

**Fig. 1 | Public opinion across 23 EU countries regarding the conservation and management of wolves and large carnivores.** For every country (identified by a two-letter code) the distribution of answers to four questions is represented as four stacked bar plots. The top outer half circular bar plot shows the stacked posterior probabilities to answer 'strongly oppose' (−−), 'oppose' (−), 'neutral or not sure' (?), 'support' (+) or 'strongly support' (++) to the question: "Large carnivores such as wolves, brown bears and lynx have been recolonizing parts of Europe in recent decades. Generally speaking, would you say that you support or oppose the recovery of large carnivores?" The top inner half circular bar plot shows the stacked posterior probabilities to answer 'decreased greatly' (−−), 'decreased' (−), 'I don't know' (?), 'stay about the same' (=), 'increased' (+) or 'increased greatly' (++) to the statement: "In my opinion, wolf populations in Europe should be...". The bottom outer half circular bar plot shows the stacked posterior probabilities to answer 'strongly oppose' (−−), 'oppose' (−), 'neutral or not sure' (?), 'support' (+) or 'strongly support' (++) to the question: "Generally speaking, would you say that you support or oppose hunting large carnivores?". The bottom inner half circular bar plot shows the stacked posterior probabilities to answer 'strongly disagree' (−−), 'somewhat disagree' (−), 'neither agree nor disagree' (?), 'somewhat agree' (+) or 'strongly agree' (++) to the statement: "Wolves that kill livestock should be killed.". Country codes: AT, Austria; BE, Belgium; BG, Bulgaria; CZ, Czechia; DE, Germany; DK, Denmark; EE, Estonia; ES, Spain; FI, Finland; FR, France; GR, Greece; HR, Croatia; HU, Hungary; IT, Italy; LT, Lithuania; LV, Latvia; NL, Netherlands; PL, Poland; PT, Portugal; RO, Romania; SE, Sweden; SI, Slovenia; SK, Slovakia. Map created with Natural Earth (https://www.naturalearthdata.com).

respondents in Italy, Spain and Bulgaria—home to some of Europe's largest wolf populations and, in the case of Bulgaria, substantial numbers of brown bears—still express majority support for large carnivore recovery. Even in countries where large carnivores cause tangible damages, such as the >10,000 sheep killed by wolves annually in France, public support for predators remains strong. However, and crucially, this support for recovery does not necessarily translate into support for population increases. On the contrary, Europeans are generally favourable to the presence of large carnivores but not to further growth in their numbers. This pattern is particularly clear for wolves: in all surveyed countries except Spain and Portugal, a relative majority of respondents prefer that wolf populations remain stable (Fig. 1 and Extended Data Fig. 2). Although in no country does a relative majority support a decline in wolf populations, population decreases attract more support than increases in several countries, such as Denmark and the Netherlands. More than ten countries are in favour of further increases in lynx populations (Extended Data Fig. 3) (albeit in Spain and Portugal, lynx implicitly refers to Iberian lynx *Lynx pardinus*, a species that narrowly escaped extinction and with presently less conflict than the Eurasian lynx), while only five countries express support for increasing brown bear populations (Extended Data Fig. 4). Slovakia stands out for showing a pronounced divergence in attitudes towards both species: it is the only country where a relative majority supports reducing bear numbers—possibly linked to reported bear attacks on people in recent years, yet it simultaneously displays strong support for increasing lynx populations.

Although our data indicate that Europeans generally prefer stable large carnivore populations, they also indicate Europeans tend to oppose hunting these species (Fig. 1 and Extended Data Fig. 5). Attitudes towards hunting vary by region and species, following a marked latitudinal gradient. Support for hunting is most pronounced in Scandinavia and the Baltic states, although nowhere does it constitute a majority view. By contrast, opposition to hunting is the strongest in Southern Europe, where it often commands an absolute (that is, >50%) majority and is consistent with long-standing national policies: Portugal and Italy, for example—two countries with the highest levels of opposition to hunting—have banned the practice for large carnivores for decades, in alignment with public sentiment.

When examining specific justifications for killing large carnivores outside of hunting (for example, such as killing individuals that have damaged property), a similar North–South divide emerges. Respondents in Fennoscandia and the Baltic States tend to support culling in such cases, while opposition is strongest in Mediterranean countries (Fig. 1 and Extended Data Figs. 6–8). In Italy, for example, survey participants strongly opposed culling even in response to brown bear attacks on humans. This widespread opposition may explain the protracted legal and administrative process Italian authorities undertook before authorizing the killing of a bear that fatally attacked a hiker in April 2023[6]. In contrast, Germany swiftly killed the first brown bear to return after 200 years—owing to bold behaviour rather than direct harm—possibly reflecting more permissive public attitudes towards lethal control in that country. Although attitudes towards lethal control measures vary widely across Member States—reflecting the influence of cultural and contextual factors—the results suggest greater public acceptance for targeted interventions, such as the killing of specific problem individuals, compared with broad population-level control (for example, hunting).

Strikingly, we found little difference between rural and urban residents in their opinions towards large carnivores. Support for large carnivore recovery remains strong even in rural areas. For instance, in Spain urban and rural responses diverge by only 2% regarding support for large carnivore recovery (Extended Data Fig. 1). Even in countries with lower levels of support, the rural–urban gap remains relatively narrow. Noteworthy differences are observed in only a few countries—for example, Austria, Estonia, Finland, Latvia and Romania—where

urban respondents show greater support than rural respondents (Extended Data Fig. 1 and Supplementary Table 3). These results challenge the common assumption[7] that conflict over large carnivore policies is primarily driven by an urban–rural divide (although our survey focused on rural residents and not on economic interests associated with 'rurality' such as hunting and agriculture). By contrast, demographic factors such as sex and age show stronger and more variable effects. Male respondents consistently express greater support for hunting (Supplementary Table 7). Age effects vary across countries. Older individuals consistently show greater support for a reduction of wolf populations (Supplementary Table 4). While older individuals tend to be less supportive of large carnivore recovery in most Member States, the opposite pattern emerges in Greece and Sweden (Supplementary Table 3). These exceptions suggest that, at least in some countries, younger generations may hold less favourable views—potentially complicating long-term conservation prospects.

Despite such variability, our assessment of public opinion indicates that Europe displays far less polarization—that is, division of society into two opposed and irreconcilable groups—than commonly portrayed. While a relative majority of Europeans support carnivore recovery, at least one third of the population remains neutral (Extended Data Fig. 1). In Spain, for example, 36% of respondents have no opinion, and in Slovakia, this figure rises to 59%. This pattern holds across both urban and rural areas. Notably, countries with the lowest levels of neutrality tend to express the highest overall support for large carnivores. This suggests that country-level variation in support is probably driven more by variation in the proportion of neutral respondents than variation in opposition, which appears comparatively fixed. Even on the issue of large carnivore hunting—often perceived as highly divisive—we find at least one third of respondents neither agree nor disagree (Extended Data Fig. 5). Also in contrast with some recent work in the USA[8], political identity shows only a weak relationship with attitudes towards large carnivores. Respondents who identify as politically left leaning tend to be slightly more supportive of carnivore recovery, but this effect is significant in only a few countries (Supplementary Table 3). Conversely, identifying as politically conservative is not significantly associated with opposition to large carnivores, except in Austria (Supplementary Table 3). These findings reinforce the broader pattern: support for large carnivores is not strongly driven by ideology, challenging the notion that the issue is deeply politicized at the societal level.

Across the EU, public attitudes reveal an apparent contradiction: while many respondents support maintaining stable wolf populations, they simultaneously oppose legally killing them—an action that, in most ecological contexts, may be necessary to cap population size in the absence of natural predators or strong density-dependent constraints. This reveals a potential disconnect between public preferences (no population growth) and the practical measures (hunting) that would probably be implemented to fulfil such preferences. That disconnect may, in turn, reveal a lack of public understanding of ecological systems that may be a challenge to enact policies that are both supported by the public and evidence based. The Nordic countries are a notable exception, where support for stable wolf populations occurs with a higher acceptance of hunting and culling. These findings suggest that political controversies over large carnivore management, such as wolves, are unlikely to dissipate even when the legal protection of wolves is downgraded. The core tension lies not much between opposing social groups—such as rural versus urban populations—but within the whole population itself, where many citizens express incompatible preferences: a desire to maintain stable populations while rejecting lethal interventions that may be needed to achieve that stability. Should strict legal protections be removed, political tensions will be nationalized, as decision-making authority shifts from the EU to individual Member States. While such devolution is sometimes framed as a path towards greater democratic legitimacy, national processes may not necessarily

be more responsive to public opinion. The Spanish case is illustrative: in March 2025, the national parliament authorized the removal of the wolf from the domestic list of protected species, reintroducing the hunting of wolves. Crucially, this change was enacted not as a wildlife policy but as a last-minute rider amendment to a law on reducing food waste, with the justification that wolf predation contributes to livestock losses[9]. Yet public opinion in Spain shows both strong support for wolf recovery and strong opposition to hunting, indicating that this reintroduction of hunting may be a political play (for example, ref. [10]) or reflect particular interest group preferences rather than democratic alignment, an issue at the core of contemporary conservation politics[11].

In that context, the increasingly charged political discourse surrounding large carnivores—particularly the wolves and bears—stands in stark contrast to public opinion. While parties such as the European People's Party, the largest political group in the European Parliament, have framed the wolf as a threat to rural life—invoking imagery reminiscent of Little Red Riding Hood[12]—such narratives do not reflect the broader attitudes of EU citizens. Right-wing rhetoric that symbolically links the wolf to the erosion of rural livelihoods[13] appears politically constructed rather than grounded in concerns from the broader or even conservative public. In reality, support for large carnivore recovery remains solid and not polarized across European countries—albeit contestation may still occur in local communities directly affected by these species and most of our survey respondents may never have had a direct experience of large carnivores.

Instead, the strongest point of consensus among Europeans lies in their widespread apathy to engage politically on the issue of large carnivores. In every country surveyed except Portugal, an absolute majority of respondents indicate they would not contact a politician to support predator recovery efforts—although levels of potential engagement vary across countries (Extended Data Fig. 9). In all countries, an absolute majority state they would not contact a politician to oppose such efforts (Extended Data Fig. 10). This suggests that, despite clear preferences in public opinion, large carnivore conservation remains a low-salience issue for most Europeans in terms of political mobilization. This renders policy outcomes vulnerable to regulatory capture: the technical complexity of large carnivore governance combined with low levels of public engagement creates conditions in which organized interests may disproportionately influence decision-making[14] as legal protection is weakened.

While the amendment to downgrade the legal protection status of the wolf in the Habitats Directive may not be grounded in new scientific evidence regarding species recovery[15], it appears broadly aligned with public opinion across Europe if it prevents further population increases and—importantly—does not lead to decreases. Nevertheless, this does not guarantee that subsequent policy implementation will reflect those same preferences. Without appropriate oversight, there is a risk that the consequences of downlisting—such as potential substantial population reductions—may diverge sharply from the public's support for recovery and coexistence, as the Spanish case illustrates. Although Annex V of the Habitats Directive continues to require that populations achieve and maintain favourable conservation status, there has been very little enforcement of this obligation at the EU level. The Commission's past inaction—for instance, regarding Finland, where Annex V has been interpreted as a license to exterminate the species—is cause for concern. Sweden's announced intention to halve its wolf population further underscores the potential for misuse[16]. However, wolves in Poland have always been listed in Annex V and still experienced a strong population recovery[2]. In this context, effective monitoring of wolves across Europe and science-based management will be essential to ensure that Member States do not exploit the shift in legal status to reverse decades of conservation gains. The challenge ahead lies in ensuring that legal rollback does not become a de facto license for conservation rollback—particularly when the species recovery remain well supported by the European public at large.

As the first-ever removal of strict protection of a species in Europe following a declared recovery, the downlisting of the wolf represents a policy experiment whose outcome remains uncertain. If it contributes to reduced conflicts and demonstrates that a more flexible legal framework can support coexistence, it may set a precedent for managing other recovering species than the wolf. However, if it results in conservation backsliding or renewed controversy, reinstating strict protection may become necessary—although such a reversal would not be automatic and would require political momentum contrary to that which enabled the downlisting.

## Methods

### Opinion survey

We conducted an opinion survey between 2022 and 2023 of 10,807 respondents across all 23 European countries that host breeding populations of large carnivores, using the Qualtrics online survey platform (www.qualtrics.com). For each surveyed country, we aimed to include at least 500 respondents, with a balanced distribution across key demographic categories: an equal number of female and male participants, equal representation of individuals younger and older than the country-specific median age (Supplementary Table 1) and an equal number of respondents from rural and urban areas. This sampling strategy was designed to avoid underrepresentation of demographics that are both harder to reach and potentially influential in shaping large carnivore policy, such as older individuals in rural regions. Respondents were asked a series of questions on wildlife and environmental issues, among which were questions to assess their attitudes towards large carnivore recovery in Europe; their views on desirable future population trends for wolves, lynx and brown bears; and their opinions on the hunting and culling of these species. In addition, we collected demographic and political covariates. We provide the questions used in this article and their translations ('Supplementary Survey Translations' in Supplementary Information).

### Bayesian models of ordinal data

We modelled the survey responses as ordinal variables, assuming they arise from an underlying normally distributed latent variable that is partitioned into categories by discrete threshold values[17]. This approach avoids the common but problematic practice of treating ordinal data as if it was metric, which can introduce systematic errors—including inflated type I and II error rates and effect reversals[18]. Metric models incorrectly assume equal intervals between ordinal categories, leading to potential misinterpretation, for example, that a response of 'strongly disagree' reflects twice the intensity of opinion as 'disagree.' Ordinal or ordered-probit models, by contrast, avoid this incorrect assumption. We used Bayesian models because frequentist ordered-probit approaches rely on optimization algorithms to estimate maximum likelihood parameters that can struggle with convergence, may fail to identify global maxima and often yield overly optimistic $P$ values and confidence intervals, particularly when sample sizes are small to moderate[18].

For each survey question, we implemented two distinct models based on the ordinal modelling framework described above (also shown in the published source code). The first was a group-based model, which estimated the parameters of the underlying probability distributions and threshold values separately for two groups: respondents living in rural areas and those in urban areas. The second model incorporated linear predictors applied to the latent continuous variable, allowing us to assess the influence of individual-level covariates on response tendencies. In both models, we treated respondents' answers as realizations of a latent normally distributed variable with mean $\mu$ and standard deviation $\sigma$ and a set of threshold values $\theta_1, \theta_2, \dots, \theta_{K-1}$ delineating the $K$ ordinal categories corresponding to the possible response options. While the latent variable is assumed to follow a normal distribution, the resulting ordinal outcomes do not

necessarily exhibit a normal distribution. The probability of a specific ordinal response corresponds to the area under the normal curve bounded by the thresholds for that category. For a given outcome $k$, the probability $p(y = k|\mu, \sigma, \{\theta_j\})$ of an answer falling into a particular category $k$ was computed as the difference between the cumulative distribution function of the normal distribution evaluated at adjacent thresholds. Formally, the probability of observing outcome $k$ given parameters $\mu$, $\sigma$ and thresholds $\theta_j$ was

$$p(y = k|\mu, \sigma, \{\theta_j\}) = \Phi\left(\frac{\theta_k - \mu}{\sigma}\right) - \Phi\left(\frac{\theta_{k-1} - \mu}{\sigma}\right),$$

where $\Phi$ denotes the cumulative distribution function of the standard normal distribution. To compute the probabilities for the extreme categories, the model incorporated virtual thresholds at $-\infty$ and $+\infty$. For questions with 'I don't know' answers, we separately estimated the probability to give such answer rather than treating it as an ordinal category. We specified weakly informative priors for $\mu$ and $\sigma$ to reflect the absence of strong prior knowledge. The mean $\mu$ was assigned a normal prior centred at the midpoint of the ordinal scale, with a wide variance. The standard deviation $\sigma$ was given a uniform prior over a broad range, allowing for flexibility in the dispersion of the latent variable. For the thresholds $\theta_j$, we used normal priors centred at $k + 0.5$ and with a wide variance. We did not assign priors to the lowest and highest thresholds $\theta_1$ and $\theta_{K-1}$; instead, these were fixed based on the bounds of the ordinal response scale. All models were implemented within a Bayesian framework using Markov Chain Monte Carlo sampling to approximate the posterior distributions of the parameters, using six chains and convergence was assessed using the Gelman–Rubin diagnostic statistic[19]. All analyses were conducted using the R statistical software v. 4.5.0 (ref. 20) with JAGS 4.3.2 (ref. 21).

### Group-based models and country-weighted averaging

As our survey panels were constructed to include equal numbers of rural and urban respondents—an approach that does not reflect the actual population distribution within each country—we estimated separate parameters for rural and urban respondents before computing the weighted averages to derive country-level estimates. The group-based model therefore had group-specific parameters: means $\mu_R$, $\mu_U$ and standard deviations, $\sigma_R$, $\sigma_U$ for rural (R) and urban (U) respondents, respectively. Modelling the two groups separately was necessary to compute valid country-level estimates; a model that did not differentiate between rural and urban respondents would not permit generalization at the national scale. This framework also allowed us to estimate differences in the distributions of ordinal scores between the two groups. The thresholds defining the ordinal categories $\theta_1, \theta_2, \ldots, \theta_{K-1}$ were held constant across groups, as the response categories were uniformly defined regardless of group membership. For a given observation belonging to group $g \in \{R, U\}$, the probability of observing a particular ordinal category $k$ was conditional on the group-specific mean $\mu_g$ and standard deviation $\sigma_g$. Formally, for each group $g$

$$p(y = k|\mu_g, \sigma_g, \{\theta_j\}) = \Phi\left(\frac{\theta_k - \mu_g}{\sigma_g}\right) - \Phi\left(\frac{\theta_{k-1} - \mu_g}{\sigma_g}\right).$$

Weakly informative priors were also assigned to the group-specific means $\mu_R$, $\mu_U$ and standard deviations $\sigma_R$, $\sigma_U$, reflecting the absence of prior assumptions about differences between rural and urban populations. The shared thresholds $\theta_j$ were given normal priors centred at the midpoints of the ordinal categories. To derive country-level estimates, we combined posterior samples from the rural and urban distributions of the latent variable, weighting them according to the actual proportions of rural and urban populations in each country (Supplementary Table 2).

### Models with linear predictors

The second model incorporated linear predictors by modifying the computation of the latent variable's mean to reflect the influence of covariates. The fundamental approach of modelling ordinal responses through cumulative normal distributions and thresholding remained unchanged. By including explanatory covariates, the model allowed us to investigate the underlying factors shaping attitudes towards large carnivores. The covariates included location (categorical: rural versus urban), sex (categorical: female versus male), age (continuous) and political orientation (categorical: extreme left, left, right, extreme right; each compared with the political centre). To incorporate these predictors into the ordinal framework, we extended the Bayesian model by expressing the mean of the latent variable, $\mu$, as a linear function of the covariates

$$\mu = \alpha + \beta_1 \text{location} + \beta_2 \text{sex} + \beta_3 \text{age} + \beta_4 \text{politics},$$

where $\alpha$ represents the intercept and $\beta_i$ denotes the coefficients for the covariates. The standard deviation $\sigma$ was modelled independently of the predictors. Non-informative priors were assigned to both $\alpha$ and $\beta_i$, reflecting a lack of prior assumptions about the direction or magnitude of effects.

## Reporting summary

Further information on research design is available in the Nature Portfolio Reporting Summary linked to this article.

## Data availability

Survey data are available via Dryad at https://doi.org/10.5061/dryad.w6m905r2k (ref. 22).

## Code availability

Code and reproducible analysis are available at https://doi.org/10.24433/CO.3129691.v2.

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

## Acknowledgements

We acknowledge funding from the Swedish Research Council VR grants 2018-10-22 and 2020-04973 to G.C. and the Spanish Ministry of Science and Innovation grant PID2023-149634OB-I00 to J.V.L.-B.

## Author contributions

G.C., Y.E., J.T.B. and J.V.L.-B. conceived the survey and G.C. and J.T.B. carried it out. G.C. designed and carried out the data analysis. G.C. wrote the paper together with Y.E., J.T.B. and J.V.L.-B.

## Funding

## Competing interests

The study was approved the Swedish Ethical Review Authority (permit no. 2022-04276-01). The authors report competing interests: G.C. is a member of the International Union for Conservation of Nature (IUCN) Species Survival Commission (SSC) Large Carnivore Initiative for Europe (LCIE), of the French government–appointed Scientific Council for Wolf and Pastoralism and of the French government–appointed Scientific Council for the Lynx National Action Plan, and a member of the Bern Convention Group of Experts on Large Carnivores at the Council of Europe, all unpaid advisory roles. J.V.L.-B. is a member of the Canid Specialist Group of the IUCN SSC and a member of the Scientific Committee advising the Spanish Ministry for the Ecological Transition and the Demographic Challenge, both unpaid advisory roles. J.T.B. is a member of the scientific advisory boards of Project Coyote and the International Wildlife Coexistence Council, both unpaid advisory roles, and is currently working in a paid capacity with Panthera, an international organization that advocates for wild felid conservation. The remaining author (Y.E.) declares no competing interests.

## Additional information

**Extended data** is available for this paper at https://doi.org/10.1038/s41559-025-02914-1.

**Correspondence and requests for materials** should be addressed to Guillaume Chapron.

**Do you support or oppose the recovery of large carnivores?**

**Extended Data Fig. 1 | Structure of European public opinion regarding the recovery of large carnivores.** Left barplot: For every country, we show the stacked posterior probabilities (in %) to answer 'Strongly oppose', 'Oppose', 'Neutral or not sure', 'Support' or 'Strongly support' to the question: '*Large carnivores such as wolves, brown bears and lynx have been recolonizing parts of Europe in recent decades. Generally speaking, would you say that you support or oppose the recovery of large carnivores?*' (Q1). Relative majorities of 'Neutral or not sure' answers are shown in darker grey. Right table: For every country, we show separately for rural (R) and urban (U) residents, the sum of the posterior probabilities (in %) to answer 'Strongly oppose' and 'Oppose' (−) and answer 'Support' or 'Strongly support' (+), with color shades as visual guides. On the right side, we show the divergence between the posterior probabilities for rural and urban residents, with color shades as visual guides (0 means perfect overlap between the two posteriors, while 100 means no overlap at all).

**Wolf populations in Europe should be:**

Legend: I don't know | Decreased greatly | Decreased | Stay about the same | Increased | Increased greatly

Left barplot data (by country): I don't know, Stay about the same, Increased, Increased greatly:

| Country | I don't know | Decreased greatly | Decreased | Stay about the same | Increased | Increased greatly |
|---|---|---|---|---|---|---|
| PT | 14 | | | 34 | 37 | 11 |
| ES | 9 | | 7 | 40 | 26 | 16 |
| AT | 9 | | 9 | 43 | 27 | 8 |
| CZ | 7 | | 7 | 49 | 27 | 7 |
| HR | 11 | | 8 | 46 | 25 | 8 |
| BE | 13 | | 8 | 42 | 25 | 8 |
| GR | 15 | | | 46 | 22 | 10 |
| PL | 11 | | 10 | 45 | 22 | 9 |
| RO | 9 | | 7 | 49 | 23 | 8 |
| FR | 8 | | 8 | 50 | 21 | 10 |
| DE | 7 | 6 | 14 | 43 | 23 | 7 |
| HU | 8 | | | 53 | 24 | 6 |
| SE | 12 | | 11 | 44 | 20 | 9 |
| BG | 14 | 7 | 9 | 44 | 20 | 7 |
| SK | 10 | 6 | 13 | 47 | 17 | 8 |
| IT | 9 | | 9 | 55 | 17 | 7 |
| DK | 8 | 12 | 14 | 43 | 17 | 7 |
| NL | 10 | 14 | 18 | 34 | 18 | 6 |
| FI | 10 | 5 | 14 | 48 | 17 | 6 |
| SI | 8 | 5 | 16 | 51 | 15 | 5 |
| EE | 14 | | 13 | 49 | 15 | |
| LT | 12 | 8 | 17 | 50 | 11 | |
| LV | 24 | | 13 | 49 | 8 | |

Right table:

| | R− | R+ | U− | U+ | % diff |
|---|---|---|---|---|---|
| PT | 7 | 51 | 4 | 57 | 5 |
| ES | 9 | 50 | 10 | 46 | 4 |
| AT | 18 | 36 | 12 | 40 | 7 |
| CZ | 11 | 37 | 10 | 37 | 2 |
| HR | 14 | 35 | 9 | 39 | 6 |
| BE | 15 | 37 | 13 | 38 | 2 |
| GR | 14 | 33 | 6 | 39 | 12 |
| PL | 18 | 30 | 14 | 39 | 10 |
| RO | 15 | 33 | 10 | 34 | 8 |
| FR | 16 | 31 | 11 | 35 | 7 |
| DE | 27 | 31 | 19 | 33 | 10 |
| HU | 11 | 29 | 9 | 35 | 5 |
| SE | 24 | 27 | 14 | 36 | 12 |
| BG | 22 | 28 | 15 | 33 | 8 |
| SK | 21 | 25 | 21 | 30 | 6 |
| IT | 15 | 21 | 12 | 28 | 9 |
| DK | 28 | 27 | 28 | 25 | 1 |
| NL | 40 | 22 | 35 | 27 | 6 |
| FI | 23 | 23 | 21 | 27 | 5 |
| SI | 27 | 21 | 20 | 21 | 8 |
| EE | 27 | 15 | 17 | 26 | 15 |
| LT | 32 | 13 | 27 | 16 | 6 |
| LV | 21 | 15 | 22 | 14 | 2 |

**Extended Data Fig. 2 | Structure of European public opinion regarding the future of wolf populations.** Left barplot: For every country, we show stacked posterior probabilities (in %) to answer 'I don't know' 'Decreased greatly', 'Decreased', 'Stay about the same', 'Increased' or 'Increased greatly' to the statement: '*In my opinion, wolf populations in Europe should be...*' (Q2). Relative majorities of 'Stay about the same' answers are shown in darker shade. Right table: For every country, we show separately for rural (R) and urban (U) residents, the sum of the posterior probabilities (in %) to answer 'Decreased greatly' and 'Decreased' (−) and answer 'Increased' or 'Increased greatly' (+), given not answering 'I don't know', and with color shades as visual guides. On the right side, we show the divergence between the posterior probabilities for rural and urban residents, with color shades as visual guides (0 means perfect overlap between the two posteriors, while 100 means no overlap at all).

**Lynx populations in Europe should be:**

Legend: I don't know | Decreased greatly | Decreased | Stay about the same | Increased | Increased greatly

| | R− | R+ | U− | U+ | % diff |
|---|---|---|---|---|---|
| ES | 3 | 73 | 5 | 75 | 7 |
| PT | 5 | 68 | 3 | 72 | 5 |
| HR | 10 | 48 | 6 | 60 | 14 |
| AT | 10 | 52 | 10 | 51 | 3 |
| SK | 9 | 48 | 8 | 56 | 9 |
| PL | 8 | 53 | 8 | 51 | 2 |
| CZ | 5 | 51 | 5 | 47 | 6 |
| SI | 12 | 49 | 10 | 44 | 8 |
| RO | 15 | 41 | 9 | 47 | 9 |
| EE | 9 | 38 | 7 | 51 | 12 |
| LV | 8 | 45 | 10 | 50 | 10 |
| BG | 20 | 38 | 9 | 48 | 14 |
| SE | 11 | 42 | 9 | 42 | 3 |
| FI | 9 | 39 | 11 | 42 | 5 |
| FR | 14 | 37 | 9 | 43 | 8 |
| DE | 10 | 44 | 12 | 38 | 6 |
| BE | 16 | 39 | 14 | 40 | 2 |
| LT | 14 | 37 | 12 | 36 | 4 |
| IT | 11 | 30 | 10 | 36 | 5 |
| GR | 14 | 34 | 8 | 38 | 10 |
| HU | 13 | 29 | 11 | 32 | 4 |
| NL | 36 | 24 | 30 | 32 | 9 |
| DK | 22 | 26 | 23 | 26 | 2 |

**Extended Data Fig. 3 | Structure of European public opinion regarding the future of lynx populations.** <u>Left barplot:</u> For every country, we show stacked posterior probabilities (in %) to answer 'I don't know' 'Decreased greatly', 'Decreased', 'Stay about the same', 'Increased' or 'Increased greatly' to the statement: '*In my opinion, lynx populations in Europe should be…*' (Q3). Relative majorities of 'Stay about the same' answers are shown in darker shade. <u>Right table:</u> For every country, we show separately for rural (R) and urban (U) residents, the sum of the posterior probabilities (in %) to answer 'Decreased greatly' and 'Decreased' (−) and answer 'Increased' or 'Increased greatly' (+), given not answering 'I don't know', and with color shades as visual guides. On the right side, we show the divergence between the posterior probabilities for rural and urban residents, with color shades as visual guides (0 means perfect overlap between the two posteriors, while 100 means no overlap at all).

**Bear populations in Europe should be:**

Legend: I don't know | Decreased greatly | Decreased | Stay about the same | Increased | Increased greatly

| Country | I don't know | Decreased greatly / Decreased | Stay about the same | Increased | Increased greatly |
|---|---|---|---|---|---|
| ES | 7 | | 27 | 37 | 25 |
| HR | 9 | | 38 | 39 | 10 |
| GR | 12 | | 35 | 30 | 17 |
| PT | 13 | | 36 | 36 | 10 |
| BG | 11 | | 40 | 32 | 9 |
| PL | 11 | 5 | 43 | 29 | 10 |
| RO | 6 | 9 | 43 | 24 | 14 |
| AT | 9 | 6 | 45 | 29 | 8 |
| FR | 10 | 6 | 45 | 25 | 11 |
| SE | 12 | 7 | 48 | 21 | 9 |
| BE | 15 | 8 | 43 | 22 | 7 |
| DE | 8 | 11 | 47 | 22 | 7 |
| CZ | 8 | 8 | 52 | 24 | 5 |
| LT | 15 | 7 | 45 | 22 | 7 |
| IT | 8 | 10 | 52 | 19 | 8 |
| HU | 8 | 8 | 52 | 21 | 6 |
| EE | 12 | 7 | 51 | 21 | 5 |
| FI | 11 | 7 | 52 | 20 | 7 |
| DK | 11 | 10 | 49 | 19 | 6 |
| NL | 12 | 11 / 14 | 40 | 17 | 6 |
| LV | 19 | 7 | 50 | 16 | 6 |
| SI | 5 | 6 / 22 | 49 | 14 | |
| SK | 8 | 14 / 28 | 35 | 10 | |

| | R− | R+ | U− | U+ | % diff |
|---|---|---|---|---|---|
| ES | 3 | 67 | 5 | 66 | 10 |
| HR | 4 | 53 | 5 | 55 | 9 |
| GR | 10 | 49 | 7 | 55 | 6 |
| PT | 6 | 52 | 6 | 54 | 3 |
| BG | 14 | 36 | 7 | 51 | 15 |
| PL | 8 | 41 | 8 | 45 | 5 |
| RO | 13 | 44 | 16 | 38 | 5 |
| AT | 12 | 37 | 9 | 43 | 7 |
| FR | 11 | 36 | 10 | 43 | 7 |
| SE | 14 | 32 | 9 | 36 | 8 |
| BE | 15 | 34 | 14 | 35 | 2 |
| DE | 20 | 30 | 16 | 33 | 4 |
| CZ | 13 | 35 | 11 | 30 | 7 |
| LT | 16 | 27 | 12 | 37 | 10 |
| IT | 17 | 24 | 12 | 31 | 8 |
| HU | 16 | 24 | 14 | 32 | 8 |
| EE | 16 | 23 | 8 | 34 | 15 |
| FI | 11 | 29 | 12 | 30 | 3 |
| DK | 16 | 27 | 16 | 29 | 3 |
| NL | 31 | 22 | 28 | 27 | 5 |
| LV | 7 | 28 | 13 | 26 | 10 |
| SI | 33 | 17 | 25 | 21 | 9 |
| SK | 48 | 14 | 44 | 18 | 4 |

**Extended Data Fig. 4 | Structure of European public opinion regarding the future of bear populations.** <u>Left barplot:</u> For every country, we show stacked posterior probabilities (in %) to answer 'I don't know' 'Decreased greatly', 'Decreased', 'Stay about the same', 'Increased' or 'Increased greatly' to the statement: '*In my opinion, brown bear populations in Europe should be…*' (Q4). Relative majorities of 'Stay about the same' answers are shown in darker shade. <u>Right table:</u> For every country, we show separately for rural (R) and urban (U) residents, the sum of the posterior probabilities (in %) to answer 'Decreased greatly' and 'Decreased' (−) and answer 'Increased' or 'Increased greatly' (+), given not answering 'I don't know', and with color shades as visual guides. On the right side, we show the divergence between the posterior probabilities for rural and urban residents, with color shades as visual guides (0 means perfect overlap between the two posteriors, while 100 means no overlap at all).

## Do you support or oppose hunting large carnivores?

Legend: Strongly oppose | Oppose | Neutral or not sure | Support | Strongly support

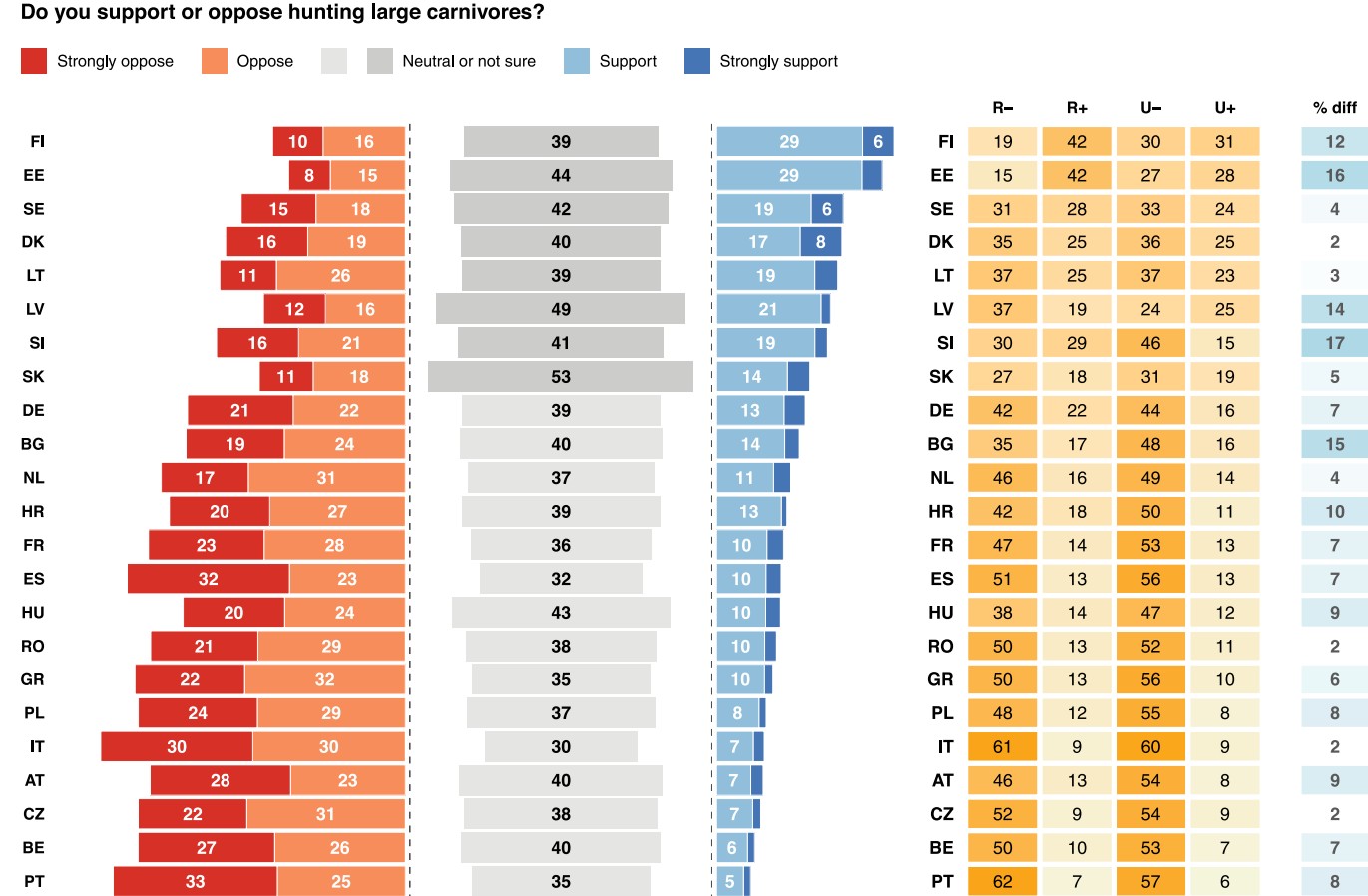

| | R− | R+ | U− | U+ | % diff |
|---|---|---|---|---|---|
| FI | 19 | 42 | 30 | 31 | 12 |
| EE | 15 | 42 | 27 | 28 | 16 |
| SE | 31 | 28 | 33 | 24 | 4 |
| DK | 35 | 25 | 36 | 25 | 2 |
| LT | 37 | 25 | 37 | 23 | 3 |
| LV | 37 | 19 | 24 | 25 | 14 |
| SI | 30 | 29 | 46 | 15 | 17 |
| SK | 27 | 18 | 31 | 19 | 5 |
| DE | 42 | 22 | 44 | 16 | 7 |
| BG | 35 | 17 | 48 | 16 | 15 |
| NL | 46 | 16 | 49 | 14 | 4 |
| HR | 42 | 18 | 50 | 11 | 10 |
| FR | 47 | 14 | 53 | 13 | 7 |
| ES | 51 | 13 | 56 | 13 | 7 |
| HU | 38 | 14 | 47 | 12 | 9 |
| RO | 50 | 13 | 52 | 11 | 2 |
| GR | 50 | 13 | 56 | 10 | 6 |
| PL | 48 | 12 | 55 | 8 | 8 |
| IT | 61 | 9 | 60 | 9 | 2 |
| AT | 46 | 13 | 54 | 8 | 9 |
| CZ | 52 | 9 | 54 | 9 | 2 |
| BE | 50 | 10 | 53 | 7 | 7 |
| PT | 62 | 7 | 57 | 6 | 8 |

**Extended Data Fig. 5 | Structure of European public opinion regarding hunting large carnivores.** <u>Left barplot:</u> For every country, we show stacked posterior probabilities (in %) to answer 'Strongly oppose', 'Oppose', 'Neutral or not sure', 'Support' or 'Strongly support' to the question: *'Generally speaking, would you say that you support or oppose hunting large carnivores?'* (Q5). Relative majorities of 'Neutral or not sure' answers are shown in darker grey. <u>Right table:</u> For every country, we show separately for rural (R) and urban (U) residents, the sum of the posterior probabilities (in %) to answer 'Strongly oppose' and 'Oppose' (−) and answer 'Support' or 'Strongly support' (+), with color shades as visual guides. On the right side, we show the divergence between the posterior probabilities for rural and urban residents, with color shades as visual guides (0 means perfect overlap between the two posteriors, while 100 means no overlap at all).

**If a bear attacks a person, that bear should be killed:**

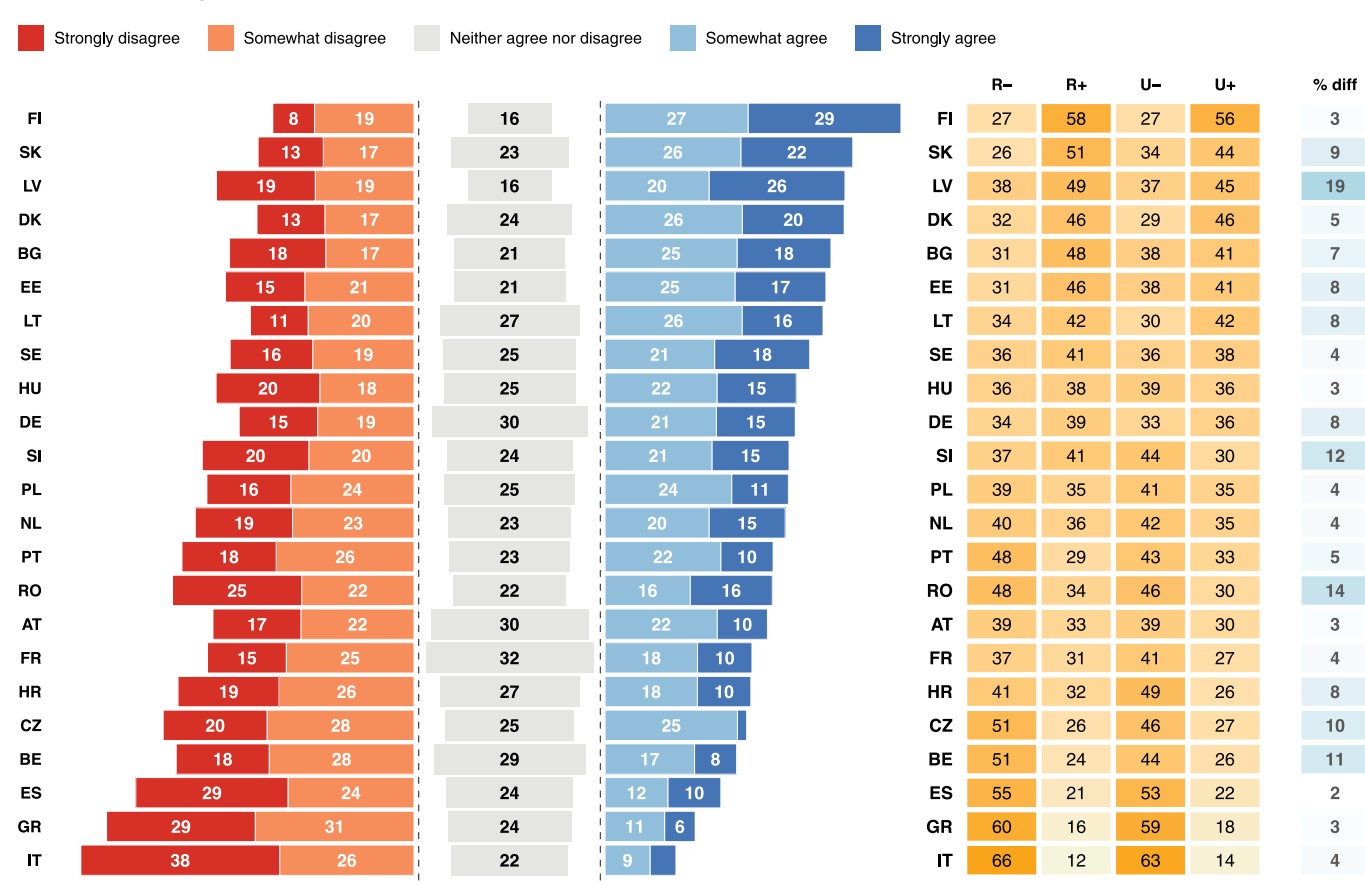

**Extended Data Fig. 6 | Structure of European public opinion regarding killing a bear that has attacked a person.** Left barplot: For every country, we show stacked posterior probabilities (in %) to answer 'Strongly disagree', 'Somewhat disagree', 'Neither agree nor disagree', 'Somewhat agree' or 'Strongly agree' to the question: '*If a bear attacks a person, that bear should be killed regardless of the circumstances*' (Q6.1). Right table: For every country, we show separately for rural (R) and urban (U) residents, the sum of the posterior probabilities (in %) to answer 'Strongly disagree' and 'Somewhat disagree' (−) and answer 'Somewhat agree' and 'Strongly agree' (+), with color shades as visual guides. On the right side, we show the divergence between the posterior probabilities for rural and urban residents, with color shades as visual guides (0 means perfect overlap between the two posteriors, while 100 means no overlap at all).

**Wolves that kill livestock should be killed:**

Legend: ■ Strongly disagree ■ Somewhat disagree ▫ Neither agree nor disagree ▪ Somewhat agree ▪ Strongly agree

| | R− | R+ | U− | U+ | % diff |
|---|---|---|---|---|---|
| FI | 19 | 63 | 26 | 53 | 10 |
| EE | 25 | 51 | 32 | 43 | 8 |
| LT | 32 | 46 | 31 | 43 | 6 |
| LV | 27 | 36 | 25 | 46 | 10 |
| DK | 33 | 42 | 32 | 42 | 3 |
| SK | 32 | 43 | 33 | 42 | 2 |
| SE | 35 | 45 | 35 | 40 | 8 |
| BG | 35 | 42 | 39 | 39 | 4 |
| SI | 37 | 41 | 42 | 35 | 6 |
| NL | 35 | 41 | 41 | 36 | 6 |
| HR | 36 | 32 | 44 | 31 | 11 |
| DE | 39 | 32 | 40 | 30 | 3 |
| HU | 43 | 32 | 45 | 29 | 2 |
| PL | 44 | 28 | 43 | 30 | 3 |
| RO | 48 | 29 | 46 | 27 | 7 |
| CZ | 49 | 27 | 47 | 24 | 8 |
| ES | 56 | 20 | 53 | 25 | 7 |
| PT | 56 | 22 | 55 | 22 | 1 |
| AT | 43 | 28 | 51 | 18 | 11 |
| FR | 46 | 22 | 49 | 23 | 5 |
| BE | 57 | 20 | 53 | 19 | 9 |
| GR | 53 | 21 | 58 | 17 | 5 |
| IT | 58 | 14 | 58 | 16 | 4 |

**Extended Data Fig. 7 | Structure of European public opinion regarding killing wolves that have killed livestock.** Left barplot: For every country, we show stacked posterior probabilities (in %) to answer 'Strongly disagree', 'Somewhat disagree', 'Neither agree nor disagree', 'Somewhat agree' or 'Strongly agree' to the question: 'Wolves that kill livestock should be killed' (Q6.2). Right table: For every country, we show separately for rural (R) and urban (U) residents, the sum of the posterior probabilities (in %) to answer 'Strongly disagree' and 'Somewhat disagree' (−) and answer 'Somewhat agree' and 'Strongly agree' (+), with color shades as visual guides. On the right side, we show the divergence between the posterior probabilities for rural and urban residents, with color shades as visual guides (0 means perfect overlap between the two posteriors, while 100 means no overlap at all).

**A lynx that kills livestock should be killed:**

Strongly disagree | Somewhat disagree | Neither agree nor disagree | Somewhat agree | Strongly agree

| | R− | R+ | U− | U+ | % diff |
|------|-----|-----|-----|-----|--------|
| FI | 24 | 55 | 29 | 49 | 7 |
| EE | 28 | 44 | 36 | 37 | 8 |
| DK | 35 | 41 | 33 | 39 | 7 |
| BG | 35 | 40 | 40 | 37 | 6 |
| LV | 29 | 33 | 31 | 40 | 14 |
| LT | 38 | 37 | 34 | 37 | 6 |
| SE | 36 | 41 | 37 | 35 | 9 |
| NL | 35 | 40 | 39 | 35 | 5 |
| SK | 37 | 34 | 40 | 34 | 4 |
| SI | 44 | 34 | 46 | 30 | 5 |
| HU | 43 | 32 | 46 | 30 | 3 |
| HR | 36 | 31 | 45 | 28 | 12 |
| DE | 43 | 30 | 43 | 28 | 4 |
| RO | 44 | 29 | 46 | 24 | 6 |
| PL | 43 | 25 | 44 | 26 | 4 |
| FR | 45 | 24 | 50 | 20 | 5 |
| PT | 58 | 20 | 58 | 19 | 2 |
| AT | 48 | 25 | 57 | 16 | 9 |
| ES | 59 | 17 | 56 | 20 | 4 |
| GR | 53 | 20 | 56 | 19 | 4 |
| BE | 55 | 20 | 52 | 19 | 8 |
| CZ | 56 | 19 | 55 | 18 | 6 |
| IT | 59 | 15 | 58 | 17 | 6 |

**Extended Data Fig. 8 | Structure of European public opinion regarding killing a lynx that has killed livestock.** Left barplot: For every country, we show stacked posterior probabilities (in %) to answer 'Strongly disagree', 'Somewhat disagree', 'Neither agree nor disagree', 'Somewhat agree' or 'Strongly agree' to the question: '*A lynx that kills livestock should be killed*' (Q6.3). Right table: For every country, we show separately for rural (R) and urban (U) residents, the sum of the posterior probabilities (in %) to answer 'Strongly disagree' and 'Somewhat disagree' (−) and answer 'Somewhat agree' and 'Strongly agree' (+), with color shades as visual guides. On the right side, we show the divergence between the posterior probabilities for rural and urban residents, with color shades as visual guides (0 means perfect overlap between the two posteriors, while 100 means no overlap at all).

**Write to or call a politician to express your support for carnivore restoration efforts:**

Legend: ■ Very unlikely ■ Somewhat unlikely □ Undecided ■ Somewhat likely ■ Very likely

| | | R− | R+ | U− | U+ | % diff |
|----|----|----|----|----|----|----|
| PT | 27 / 15 / 28 / 24 / 7 | 40 | 33 | 42 | 30 | 3 |
| BG | 25 / 26 / 25 / 22 | 45 | 28 | 54 | 21 | 10 |
| DE | 35 / 19 / 24 / 13 / 9 | 65 | 13 | 51 | 24 | 14 |
| FR | 29 / 25 / 26 / 14 / 6 | 54 | 22 | 54 | 20 | 2 |
| IT | 27 / 26 / 27 / 15 | 63 | 14 | 50 | 22 | 14 |
| PL | 29 / 25 / 26 / 16 | 61 | 14 | 51 | 24 | 11 |
| RO | 31 / 26 / 23 / 18 | 56 | 20 | 57 | 19 | 1 |
| GR | 29 / 24 / 27 / 14 / 6 | 52 | 20 | 53 | 20 | 2 |
| ES | 30 / 26 / 26 / 13 / 5 | 60 | 17 | 55 | 18 | 7 |
| BE | 33 / 26 / 24 / 12 / 5 | 63 | 16 | 57 | 17 | 7 |
| SE | 36 / 19 / 31 / 11 | 63 | 11 | 51 | 16 | 16 |
| DK | 37 / 21 / 28 / 9 | 60 | 11 | 58 | 16 | 8 |
| LT | 37 / 27 / 23 / 10 | 59 | 16 | 66 | 11 | 7 |
| NL | 43 / 23 / 22 / 8 | 69 | 10 | 66 | 13 | 3 |
| SI | 47 / 20 / 21 / 9 | 70 | 8 | 64 | 15 | 13 |
| EE | 43 / 21 / 25 / 8 | 59 | 13 | 67 | 11 | 10 |
| HU | 50 / 17 / 22 / 9 | 65 | 11 | 67 | 11 | 4 |
| AT | 46 / 24 / 19 / 7 | 76 | 7 | 67 | 12 | 11 |
| HR | 40 / 21 / 29 / 7 | 63 | 11 | 58 | 10 | 9 |
| FI | 60 / 20 / 13 / 6 | 83 | 5 | 78 | 9 | 12 |
| SK | 57 / 16 / 20 / 5 | 74 | 6 | 73 | 8 | 7 |
| CZ | 51 / 24 / 19 | 77 | 5 | 73 | 6 | 3 |
| LV | 47 / 25 / 23 | 74 | 5 | 72 | 5 | 10 |

**Extended Data Fig. 9 | Structure of European public opinion regarding contacting a politician to support large carnivore conservation.** Left barplot: For every country, we show stacked posterior probabilities (in %) to answer 'Very unlikely', 'Somewhat unlikely', 'Undecided', 'Somewhat likely' or 'Very likely' to the question: '*How likely or unlikely you are to write to or call a politician to express your support for carnivore restoration efforts?*' (Q11). Right table: For every country, we show separately for rural (R) and urban (U) residents, the sum of the posterior probabilities (in %) to answer 'Very unlikely' and 'Somewhat unlikely' (−) and answer 'Somewhat likely' and 'Very likely' (+), with color shades as visual guides. On the right side, we show the divergence between the posterior probabilities for rural and urban residents, with color shades as visual guides (0 means perfect overlap between the two posteriors, while 100 means no overlap at all).

**Write to or call a politician to express your opposition to carnivore restoration efforts:**

Legend: Very unlikely | Somewhat unlikely | Undecided | Somewhat likely | Very likely

| | R− | R+ | U− | U+ | % diff |
|---|---|---|---|---|---|
| PT | 55 | 23 | 57 | 20 | 4 |
| DE | 64 | 14 | 57 | 22 | 9 |
| IT | 73 | 10 | 60 | 18 | 13 |
| PL | 68 | 11 | 60 | 17 | 8 |
| RO | 64 | 16 | 68 | 13 | 4 |
| FR | 69 | 12 | 65 | 14 | 6 |
| DK | 63 | 15 | 63 | 13 | 5 |
| GR | 67 | 11 | 61 | 14 | 6 |
| ES | 67 | 13 | 64 | 13 | 6 |
| BG | 64 | 13 | 69 | 11 | 7 |
| LT | 67 | 12 | 67 | 11 | 3 |
| NL | 69 | 10 | 67 | 11 | 3 |
| BE | 77 | 6 | 70 | 10 | 7 |
| EE | 68 | 6 | 68 | 11 | 17 |
| HU | 72 | 8 | 71 | 10 | 10 |
| SI | 72 | 9 | 75 | 9 | 6 |
| SE | 70 | 8 | 64 | 8 | 12 |
| HR | 67 | 10 | 72 | 8 | 6 |
| AT | 82 | 7 | 82 | 6 | 12 |
| SK | 72 | 6 | 77 | 6 | 7 |
| FI | 84 | 4 | 82 | 7 | 24 |
| CZ | 81 | 5 | 84 | 3 | 8 |
| LV | 76 | 3 | 78 | 3 | 6 |

**Extended Data Fig. 10 | Structure of European public opinion regarding contacting a politician to oppose large carnivore conservation.** Left barplot: For every country, we show stacked posterior probabilities (in %) to answer 'Very unlikely', 'Somewhat unlikely', 'Undecided', 'Somewhat likely' or 'Very likely' to the question: '*How likely or unlikely you are to write to or call a politician to express your opposition for carnivore restoration efforts?*' (Q12). Right table: For every country, we show separately for rural (R) and urban (U) residents, the sum of the posterior probabilities (in %) to answer 'Very unlikely' and 'Somewhat unlikely' (−) and answer 'Somewhat likely' and 'Very likely' (+), with color shades as visual guides. On the right side, we show the divergence between the posterior probabilities for rural and urban residents, with color shades as visual guides (0 means perfect overlap between the two posteriors, while 100 means no overlap at all).

# Reporting Summary

## Statistics

For all statistical analyses, confirm that the following items are present in the figure legend, table legend, main text, or Methods section.

| n/a | Confirmed | |
|---|---|---|
| ☐ | ☒ | The exact sample size (*n*) for each experimental group/condition, given as a discrete number and unit of measurement |
| ☒ | ☐ | A statement on whether measurements were taken from distinct samples or whether the same sample was measured repeatedly |
| ☒ | ☐ | The statistical test(s) used AND whether they are one- or two-sided<br>*Only common tests should be described solely by name; describe more complex techniques in the Methods section.* |
| ☒ | ☐ | A description of all covariates tested |
| ☒ | ☐ | A description of any assumptions or corrections, such as tests of normality and adjustment for multiple comparisons |
| ☐ | ☒ | A full description of the statistical parameters including central tendency (e.g. means) or other basic estimates (e.g. regression coefficient) AND variation (e.g. standard deviation) or associated estimates of uncertainty (e.g. confidence intervals) |
| ☒ | ☐ | For null hypothesis testing, the test statistic (e.g. *F*, *t*, *r*) with confidence intervals, effect sizes, degrees of freedom and *P* value noted<br>*Give P values as exact values whenever suitable.* |
| ☐ | ☒ | For Bayesian analysis, information on the choice of priors and Markov chain Monte Carlo settings |
| ☐ | ☒ | For hierarchical and complex designs, identification of the appropriate level for tests and full reporting of outcomes |
| ☒ | ☐ | Estimates of effect sizes (e.g. Cohen's *d*, Pearson's *r*), indicating how they were calculated |

*Our web collection on statistics for biologists contains articles on many of the points above.*

## Software and code

Policy information about availability of computer code

| Data collection | No software was used (data collection was made by Qualtrics). |
|---|---|
| Data analysis | All analyses were conducted using the R statistical software v. 4.5.0 with JAGS 4.3.2. |

For manuscripts utilizing custom algorithms or software that are central to the research but not yet described in published literature, software must be made available to editors and reviewers. We strongly encourage code deposition in a community repository (e.g. GitHub). See the Nature Portfolio guidelines for submitting code & software for further information.

## Data

Policy information about availability of data

All manuscripts must include a data availability statement. This statement should provide the following information, where applicable:
- Accession codes, unique identifiers, or web links for publicly available datasets
- A description of any restrictions on data availability
- For clinical datasets or third party data, please ensure that the statement adheres to our policy

Data is available at https://doi.org/10.5061/dryad.w6m905r2k Code and reproducible analysis are available at https://doi.org/10.24433/CO.3129691.v2

# Research involving human participants, their data, or biological material

Policy information about studies with <u>human participants or human data</u>. See also policy information about <u>sex, gender (identity/presentation), and sexual orientation</u> and <u>race, ethnicity and racism</u>.

| | |
|---|---|
| Reporting on sex and gender | Participants answered the following question:<br>- What is your sex? (Male/Female) |
| Reporting on race, ethnicity, or other socially relevant groupings | Participants answered the following questions:<br>- Enter the year you were born (in YYYY format)<br>- How would you describe your current residence or community? (Large urban area / City / Village / Isolated house)<br>- When it comes to politics, please indicate which of the following you consider yourself. (Extreme left / Left (socio-democrat) / Centre / Right (conservative) / Extreme right) |
| Population characteristics | See above |
| Recruitment | Participants were recruited through Qualtrics' national panels |
| Ethics oversight | The study was approved the Swedish Ethical Review Authority (permit 2022-04276-01) |

Note that full information on the approval of the study protocol must also be provided in the manuscript.

# Field-specific reporting

Please select the one below that is the best fit for your research. If you are not sure, read the appropriate sections before making your selection.

☐ Life sciences   ☒ Behavioural & social sciences   ☐ Ecological, evolutionary & environmental sciences

For a reference copy of the document with all sections, see nature.com/documents/nr-reporting-summary-flat.pdf

# Behavioural & social sciences study design

All studies must disclose on these points even when the disclosure is negative.

| | |
|---|---|
| Study description | Quantitative opinion survey of European citizens about large carnivores. |
| Research sample | We conducted an opinion survey between 2022 and 2023 across all 23 European countries that host breeding populations of large carnivores, using the Qualtrics online survey platform (www.qualtrics.com). A total of 10,807 respondents were recruited through Qualtrics' national panels. All participants provided informed consent prior to participation, and responses were anonymized by Qualtrics before being made available to the research team. |
| Sampling strategy | For each surveyed country, we aimed to include at least 500 respondents, with a balanced distribution across key demographic categories: an equal number of female and male participants, equal representation of individuals younger and older than the country-specific median age, and an equal number of respondents from rural and urban areas (whether a respondent enters the category rural or urban is determined by Q9, with answers "Village" or "Isolated house" converting to rural, and answers "Large urban area" or "City" converting to urban). This sampling strategy was designed to avoid underrepresentation of demographics that are both harder to reach and potentially influential in shaping large carnivore policy—such as older individuals in rural regions. In a few countries, however, this targeted balance could not be fully achieved. For instance, rural respondents in Bulgaria, Croatia, and Romania were underrepresented relative to our targets. |
| Data collection | All survey responses were collected online. |
| Timing | 2022-2023 |
| Data exclusions | n/a |
| Non-participation | The authors were only given the survey responses without details on dropout rate. |
| Randomization | n/a |

# Reporting for specific materials, systems and methods

We require information from authors about some types of materials, experimental systems and methods used in many studies. Here, indicate whether each material, system or method listed is relevant to your study. If you are not sure if a list item applies to your research, read the appropriate section before selecting a response.

## Materials & experimental systems

| n/a | Involved in the study |
|-----|------------------------|
| ☒ ☐ | Antibodies |
| ☒ ☐ | Eukaryotic cell lines |
| ☒ ☐ | Palaeontology and archaeology |
| ☒ ☐ | Animals and other organisms |
| ☒ ☐ | Clinical data |
| ☒ ☐ | Dual use research of concern |
| ☒ ☐ | Plants |

## Methods

| n/a | Involved in the study |
|-----|------------------------|
| ☒ ☐ | ChIP-seq |
| ☒ ☐ | Flow cytometry |
| ☒ ☐ | MRI-based neuroimaging |

## Plants

| | |
|---|---|
| Seed stocks | *Report on the source of all seed stocks or other plant material used. If applicable, state the seed stock centre and catalogue number. If plant specimens were collected from the field, describe the collection location, date and sampling procedures.* |
| Novel plant genotypes | *Describe the methods by which all novel plant genotypes were produced. This includes those generated by transgenic approaches, gene editing, chemical/radiation-based mutagenesis and hybridization. For transgenic lines, describe the transformation method, the number of independent lines analyzed and the generation upon which experiments were performed. For gene-edited lines, describe the editor used, the endogenous sequence targeted for editing, the targeting guide RNA sequence (if applicable) and how the editor was applied.* |
| Authentication | *Describe any authentication procedures for each seed stock used or novel genotype generated. Describe any experiments used to assess the effect of a mutation and, where applicable, how potential secondary effects (e.g. second site T-DNA insertions, mosiacism, off-target gene editing) were examined.* |

