## [Peer Review File · Nature Ecology & Evolution]

Europeans support large carnivore recovery while opposing both further population growth and hunting

Corresponding Author: Dr Guillaume Chapron

Version 0:

Decision Letter:

21st July 2025

Dear Dr Chapron,

Your manuscript entitled "Rollback of Wolf Legal Protection in Europe Responds to Public Support for Coexistence" has now been seen by three reviewers, whose comments are attached. The reviewers have raised a number of concerns which will need to be addressed before we can offer publication in Nature Ecology & Evolution. We will therefore need to see your responses to these comments, along with a revised manuscript, before we can reach a final decision regarding publication.

Please could you at this stage revise your paper into our Brief Communication format, which is a shortened version of our standard Article format for research manuscripts. It will not need a lot of reformatting, but we will need the Methods moved to the end of the main text, rather than in the Supplementary Information. Ideally the abstract should be around three sentences/up to around 100 words (this is not a strict limit). It might be helpful to consider subheadings that are somewhat more 'factual' about the results.

Please see here for further information <https://www.nature.com/natecolevol/content>, and here for an example of a Brief Communication <https://www.nature.com/articles/s41559-024-02630-2>

We therefore invite you to revise your manuscript taking into account all reviewer and editor comments. Please highlight all changes in the manuscript text file.

* If you have not done so already please begin to revise your manuscript so that it conforms to our Brief Communication format instructions at <http://www.nature.com/natecolevol/info/final-submission>. Refer also to any guidelines provided in this letter.

* Extended Data Figures - please ensure that any supplementary figures and tables that are crucial to the manuscript's conclusions are converted into Extended Data figures and tables to increase visibility of these data. Extended Data figures and tables are online-only (present in the online PDF and full-text HTML versions of the paper), peer-reviewed display items that provide essential background to the article but are not included in the main article due to space constraints. A maximum of ten Extended Data display items (figures and tables) is permitted.

Link Redacted

Nature Ecology & Evolution is committed to improving transparency in authorship. As part of our efforts in this direction, we are now requesting that all authors identified as 'corresponding author' on published papers create and link their Open Researcher and Contributor Identifier (ORCID) with their account on the Manuscript Tracking System (MTS), prior to acceptance. ORCID helps the scientific community achieve unambiguous attribution of all scholarly contributions. You can create and link your ORCID from the home page of the MTS by clicking on 'Modify my Springer Nature account'. For more information please visit www.springernature.com/orcid.

[redacted]

Reviewer comments:

Reviewer #1 (Remarks to the Author):

An excellent brief communication that summarises the current challenges of wolf conservation status delisting with empirical data to highlight the conundrum of conflicting public support for population control but resistance to lethal management.

I find this manuscript a very timely and useful contribution to the debate, with a high-quality dataset that was collected and analysed, with highly interesting and policy-relevant results.

Some small comments per sections:

Introduction:

'Excluding islands, wolves are now present in all EU Member States, with many growing populations' – please provide a reference for this sentence.

'authorizing the removal of a bear' – I find this sentence mirrors the language used by administrative entities and constitutes a euphemism for lethal control. The bear was not removed to be put elsewhere, it was killed. I would encourage the authors to avoid this euphemism and instead use less euphemistic language, e.g. writing about 'lethal control' or similar, as indeed you do in the later part of the same paragraph. Suggest to apply this terminology consistently throughout.

'In reality, support for large carnivore recovery remains solid and relatively unpolarized across the European population' – can you clarify that this finding is based on your dataset? But this statement would also merit a bit more nuance, there is evidence of contestation and polarisation in the literature from more localised studies.

For instance in the media: Trainotti, N., Fedrigotti, C., Malavasi, S., Pedrini, P., & Bombieri, G. (2023). Wolf coverage and framing by newspapers across the Italian Eastern Alps. *Human Dimensions of Wildlife*, 1–18.

And also empirical evidence from small localised survey samples from studies on wolf management in the EU with people more directly affected than in your sample:

Pohja-Mykrä, M., & Kurki, S. (2014). Strong community support for illegal killing challenges wolf management. *European Journal of Wildlife Research*, 60, 759–770.

Trebo, S., Cary, E., & Wartmann, F. M. (2025). Emotions shape attitudes towards wolf conservation management in the Italian Alps. *European Journal of Wildlife Research*, 71(1), 1-16.

'The Risk of Delisting Drift' – excellent paragraph, I do wonder if you wanted to give more examples than just Spain. Another tangent, though not sure if you would consider including this case is Switzerland, though not an EU member state and therefore you did not collect any data, is following the EU directive, and has shot over 100 wolves with an estimated population of 340 after deregulating the strict protection and directly ignoring ecological scientific guidance – a cautionary tale for EU member states?

Methods:

Unfortunately, your survey contains no control variable on past direct interactions with wolves or other large carnivores or livelihoods of the respondents, which are likely strong predictor of attitudes. The closest proxy is urban vs. rural livelihood. Would you have considered including another control variable useful or was the urban/rural proxy sufficient in your opinion? I am missing a brief reflection or comment on how the survey could or should be adapted for a further iteration based on your learnings from this study in the methodology section.

Supplementary materials:

Given translatability challenges of many concepts that are not directly or easily translatable across European languages, for reproducibility reasons I would very strongly encourage the authors to share all the translated versions of their surveys in the languages in which they were distributed. This would massively aid any follow-up and more localised studies wanting to apply the same questionnaire or parts of it for comparability with future, more in-depth studies with the existing data this study has collected, without adding the confounding factor of slightly different translations used if the original translated survey versions are not readily available.

My recommendation to the editor is to accept with minor revisions as per comments above.

Reviewer #2 (Remarks to the Author):

This is a really interesting and timely paper. Although the data collection comes before the delisting of wolves it reflects the sentiment of people leading up to the decision. The text is clear and well written and the presentation of data is clear, innovative and informative. Overall I have very few issues;

- Make it explicit that in the case of Portugal and Spain it is implicit that "lynx" refers to Iberian lynx, and in the rest of Europe it refers to Eurasian lynx. Also make it explicit that these are two very different species in terms of conflict potential and conservation trajectories.
- Make it explicit that not all people sampled actually have direct experience of large carnivores, or may not even live in areas with carnivores.
- Provide more details of the Qualtrics panels and the survey - what motivates / incentivises people to be on panels? Were the questions asked on the phone or were respondents sent an online link? Discuss more about the limitations and potential biases of this form of survey across the diversity of Europe - is the profile of panel members the same across Europe (in ways not corrected for by age / sex in your study). Given the diverse social-economic-political situation and history of Europe I can imagine that there are very different degrees of willingness to engage in panels. At least discuss the issue.
- Provide details of the ethics oversights - this is not detailed clearly in the paper or supplementary materials - and on the form you only write DBPR what is that? Also a mention of compliance with GDPR rules would be useful.
- Your lack of a rural-urban difference is an interesting find - although it reflects the fact that many rural areas are occupied by people of urban origin - such that the expected difference would be between "rurality" as a form of adherence to traditional activities and with a sense of place rather than rural living per se. Again, at least mention this issue.
- The "delisting drift" section is a bit biased. The point is worth mentioning in the wider context of the interest in following this first ever delisting of a species - but it could go either way - if it really helps then it would not be a bad thing for other species to follow - but if it all goes wrong then maybe wolves could be relisted.
- It is also important to point out that your survey covers 3(4) species but the delisting is only wolves - and only changes things in a portion of countries because many others already had it on a lower protection status.
- The majority vs interest group influence is interesting - but then it might also be useful to point out that when most people are indifferent to carnivore issues it would seem fair for those interest groups most engaged to have a larger say in policy? At least discuss.
- Finally, make the argument linking the study findings and support for EU policy more clear. None of your questions asked about it explicitly - so it is implicit in the interpretation of the questions - so tidy it up a little to be clearer.

Reviewer #2 (Remarks on code availability):

My skill set does not extend to coding of bayesian statistics - so I hope that other reviewers are more code literate than I am.

I did check the link to see if it works.

Reviewer #3 (Remarks to the Author):

Overall, the article is well written, interesting, and relevant. However, there are a few aspects I would like the authors to address to improve the clarity and coherence of the manuscript.

In terms of content, some parts of the paper focus specifically on wolves, while others include bears and lynx as well, which creates a degree of confusion. I recommend either narrowing the focus exclusively to wolves or broadening the scope to consistently include all three large carnivores throughout the paper. If the latter approach is chosen, the title should be modified accordingly to reflect this broader focus.

Regarding the categorization of respondents, I understand that individuals could self-identify as living in a large urban area, city, or village. However, it would be helpful for the authors to clarify how they have defined and categorized rural versus urban areas. For instance, I would consider a village to be more suburban than rural, depending on population density.

Additionally, it is unclear whether quotas were used during sampling to ensure a representative distribution of respondents or if this was examined retrospectively. It is also worth noting, as established in previous research, that panel surveys often have a bias toward individuals with higher levels of education. This demographic skew may mean that the respondent pool is more representative of urban populations, and even if some individuals report living in rural areas, they may not accurately represent key rural groups such as shepherds, livestock owners, or farmers.

I am also interested in how the age groups were categorized—what criteria or threshold were used to define "young" versus "old"?

Finally, I would like to understand the rationale behind asking a different type of question about bears (i.e., the potential for killing humans), rather than maintaining a consistent focus on impacts such as damage to small domestic livestock (e.g., sheep, chickens, or beehives).

In the conclusions, I would like to see more direct recommendations based on the findings, rather than focusing solely on the challenges that lie ahead.

*****END*****

Version 1:

Decision Letter:

21st September 2025

Dear Dr. Chapron,

Thank you for submitting your revised manuscript "Europeans support large carnivore recovery while opposing both further population growth and hunting" (NATECOLEVOL-25061890A). It has now been seen again by two of the original reviewers, whose comments are below. Reviewer 2 was not available to re-review. The reviewers find that the paper has improved in revision, and therefore we'll be happy in principle to publish it in Nature Ecology & Evolution, pending minor revisions to satisfy the reviewers' final suggestions and to comply with our editorial and formatting guidelines.

If you have not done so already, please ensure that you also email us a completed copy of the Reporting summary :

Reporting summary: https://www.nature.com/documents/nr-reporting-summary.pdf

[redacted]

Reviewer #1 (Remarks to the Author):

Thank you for the revisions, I think the manuscript has improved in readability and precision of the argument.

I have two very minor comments:

The abstract does not list any other carnivores than wolves. Given that you decided to stay with the title of carnivores after reviewer comments, I would suggest to also name European lynx and brown bear somewhere in the abstract.

P.3 line 88 - you mention 'compatriots' but I don't think this is actually correct. The panels do not typically distinguish between residents and nationals. Unless you specifically selected only nationals of a sampled country, please replace this with another term that does not indicate nationality.

Other than that I have no further comments and look forward to seeing this manuscript published.

Reviewer #3 (Remarks to the Author):

While I believe the article is largely ready, I have not yet received full answers to some of my previous comments. I may not have explained myself clearly before, so let me clarify:

Rural–urban definition:

I understand that each country has its own criteria for defining what is urban and what is rural, which makes sense. However, I think I did not explain myself clearly earlier: How exactly did you divide the sample for the rural–urban analysis, based on the question included in the questionnaire?

Panel issues and potential bias in urban–rural differences:

Again, I may not have been clear before. My comment was about possible panel bias, specifically that rural respondents

who participate in panels often tend to have a higher level of education than the general rural population. Did you control for this factor? In other studies, researchers have found little to no difference between urban and rural responses due to this effect. I believe it would be important to add a limitation regarding the use of panels. For example, see Martínez-Jauregui et al. (2023): <https://doi.org/10.1016/j.jenvman.2023.117236>

Age definition:

What is the threshold for being considered “young”? I still do not understand how the authors divided the sample into “young” and “old” groups based on the categorical question.

Responses to reviewers NATECOLEVOL-25061890

Reviewer #1:

Introduction:

'Excluding islands, wolves are now present in all EU Member States, with many growing populations' – please provide a reference for this sentence.

This sentence was not essential and removed when shortening the manuscript to fit the Brief Communications format.

'authorizing the removal of a bear' – I find this sentence mirrors the language used by administrative entities and constitutes a euphemism for lethal control. The bear was not removed to be put elsewhere, it was killed. I would encourage the authors to avoid this euphemism and instead use less euphemistic language, e.g. writing about 'lethal control' or similar, as indeed you do in the later part of the same paragraph. Suggest to apply this terminology consistently throughout.

We have replaced 'removal' with 'killing', also in other instances in the manuscript.

'In reality, support for large carnivore recovery remains solid and relatively unpolarized across the European population' – can you clarify that this finding is based on your dataset?

We explain what we mean at lines 104–107, where we wrote “*Despite such variability, our assessment of public opinion indicates Europe displays far less polarization—i.e., division of society into two opposed and irreconcilable groups—than commonly portrayed. While a relative majority of Europeans support carnivore recovery, at least one third of the population remains neutral (Extended Data Fig. 1)*”. The reference to the Extended Data figure makes the direct link with the data.

But this statement would also merit a bit more nuance, there is evidence of contestation and polarisation in the literature from more localised studies. For instance in the media: Trainotti, N., Fedrigotti, C., Malavasi, S., Pedrini, P., & Bombieri, G. (2023). Wolf coverage and framing by newspapers across the Italian Eastern Alps. *Human Dimensions of Wildlife*, 1–18. And also empirical evidence from small localised survey samples from studies on wolf management in the EU with people more directly affected than in your sample: Pohja-Mykrä, M., & Kurki, S. (2014). Strong community support for illegal killing challenges wolf management. *European Journal of Wildlife Research*, 60, 759–770. Trebo, S., Cary, E., & Wartmann, F. M. (2025). Emotions shape attitudes towards wolf conservation management in the Italian Alps. *European Journal of Wildlife Research*, 71(1), 1-16.

We appreciate the reviewer's concern; but it seems that the reviewer treats contestation and polarisation similarly. However we write only about polarisation, and not contestation. The

Merriam-Webster's online dictionary defines polarization as “division into two sharply distinct opposites; especially, a state in which the opinions, beliefs, or interests of a group or society no longer range along a continuum but become concentrated at opposing extremes.” Our opinion survey does not find this pattern (concentration at the extremes is lacking), neither do the suggested references (focusing on contestation). We do not mean that large carnivore and related policies are un-contested (our results indeed show that some opposition can be found in every country). Importantly, we focus on public opinion and not on media reporting. This is important because the media discourse on wolves is effectively polarizing (see e.g. Zscheischler, J., & Friedrich, J. (2022). The wolf (*Canis lupus*) as a symbol of an urban–rural divide? Results from a media discourse analysis on the human–wolf conflict in Germany. *Environmental Management*, 70(6), 1051-1065). However, the presentation of an issue as 'polarized' in the news media is not an indication that polarization actually exists in society. For example, Bruskotter et al (2018) demonstrated that despite numerous examples of the Endangered Species Act being presented as controversial in the news media, polls consistently indicate the opposite (Bruskotter, J. T., Vucetich, J. A., Slagle, K. M., Berardo, R., Singh, A. S., & Wilson, R. S. (2018). Support for the US Endangered Species Act over time and space: Controversial species do not weaken public support for protective legislation. *Conservation Letters*, 11(6), e12595.).

To make our message clearer, we have modified our text as follows. First, we have clarified what is polarization: “*Despite such variability, our assessment of public opinion indicates it displays far less polarization –or sharp division of society into two opposed and irreconcilable groups– than commonly portrayed*” (line 105). And we have also modified this sentence: “*In reality, support for large carnivore recovery remains solid and not polarized across European countries (albeit contestation may still occur in local communities directly affected by these species).*” (lines 158–159).

‘The Risk of Delisting Drift’ – excellent paragraph, I do wonder if you wanted to give more examples than just Spain. Another tangent, though not sure if you would consider including this case is Switzerland, though not an EU member state and therefore you did not collect any data, is following the EU directive, and has shot over 100 wolves with an estimated population of 340 after deregulating the strict protection and directly ignoring ecological scientific guidance – a cautionary tale for EU member states?

Switzerland is an interesting example, as would Norway, but because these countries are not in the EU (and were not sampled), we could be told of using examples from different contexts to make a particular point. Also, reviewer 2 suggests to bring nuance to this paragraph.

Methods:

Unfortunately, your survey contains no control variable on past direct interactions with wolves or other large carnivores or livelihoods of the respondents, which are likely strong predictor of attitudes. The closest proxy is urban vs. rural livelihood. Would you have considered including another control variable useful or was the urban/rural proxy sufficient in your opinion? I am missing a brief reflection or comment on how the survey could or should be

adapted for a further iteration based on your learnings from this study in the methodology section.

Considering that large carnivores occur at low densities, in areas where human population densities are also low, and that our sample size was typically 500 respondents per country, we would anticipate that extremely few individuals in our sample would have directly interacted with a large carnivore and this would render such a variable non-informative in our analysis. In addition, our intent was not to attempt to explain attitudes at the individual level but rather to describe public opinion at the country level across the EU. The reason we forced a quota of rural respondents was to avoid missing demographics that are both harder to reach and potentially influential in shaping large carnivore policy.

Supplementary materials:

Given translatability challenges of many concepts that are not directly or easily translatable across European languages, for reproducibility reasons I would very strongly encourage the authors to share all the translated versions of their surveys in the languages in which they were distributed. This would massively aid any follow-up and more localised studies wanting to apply the same questionnaire or parts of it for comparability with future, more in-depth studies with the existing data this study has collected, without adding the confounding factor of slightly different translations used if the original translated survey versions are not readily available.

We have now put the translated questionnaires at the end of the supplementary material.

Reviewer #2:

Make it explicit that in the case of Portugal and Spain it is implicit that "lynx" refers to Iberian lynx, and in the rest of Europe it refers to Eurasian lynx. Also make it explicit that these are two very different species in terms of conflict potential and conservation trajectories.

We have added: "*albeit in Spain and Portugal, lynx implicitly refers to Iberian lynx Lynx pardinus, a species which narrowly escaped extinction and with presently less conflict than the Eurasian lynx*" lines 58–59

Make it explicit that not all people sampled actually have direct experience of large carnivores, or may not even live in areas with carnivores.

We add line 159 "*most of the respondents may never had a direct experience of large carnivores*"

Provide more details of the Qualtrics panels and the survey - what motivates / incentivises people to be on panels? Were the questions asked on the phone or were respondents sent

an online link? Discuss more about the limitations and potential biases of this form of survey across the diversity of Europe - is the profile of panel members the same across Europe (in ways not corrected for by age / sex in your study). Given the diverse social-economic-political situation and history of Europe I can imagine that there are very different degrees of willingness to engage in panels. At least discuss the issue.

We now discuss the issue in more details. We added a section “**Panelist recruitment**” in the supplementary methods (since it would take too much space in the main text).

Panelist recruitment

Qualtrics uses several sources for its panel aggregation (e.g., targeted emails, website intercepts, social media), and places caps on how many can come from a specific recruitment source to prevent a bias from entering into data. Each of the individual panelists are vetted further by a proprietary technology which performs a digital fingerprint check to ensure people are who they say they are. Panelists are then invited to participate in surveys via email informing them that they will be compensated (the type and extent of compensation may vary). Qualtrics applies additional quality control measures to exclude responses from participants who completed the survey too quickly or exhibited patterned or non-serious answering behavior.

Studies generally indicate this type of non-probability sampling has higher error rates when compared with known benchmarks (e.g., Lehdonvirta et al. 2021). However, a recent study focusing on wildlife-related issues found that a non-probability Qualtrics sample out-performed a probability-based phone survey in terms of replicating benchmarks (Vaske et al. 2022). This study also showed that mail, phone, and internet-based surveys were all biased toward those who participated in wildlife-related activities (e.g., hunting, fishing), which likely drives greater interest in wildlife-related issues. Thus, it may not be unlikely that our data likewise over-represent people who have some interest in wildlife issues. The fact that our panels do not show polarization or willingness to contact politicians give us confidence about the generalization of our results at the country population level.

European nations being diverse in terms of their social and economic conditions, these conditions may affect the propensity of individuals to respond. For example, Daikeler et al. 2023 showed that country-level response rates to web-based surveys tend to be higher (relative to traditional modes of contact; e.g., mailed survey) in countries with high population growth, high internet coverage, and lower (again relative to traditional modes) in countries with a high population age and high cell phone coverage. We mitigated for this possible bias by specifically requesting a quota of rural respondents.

Provide details of the ethics oversights - this is not detailed clearly in the paper or supplementary materials - and on the form you only write DBPR what is that? Also a mention of compliance with GDPR rules would be useful.

DBPR stands for double blind peer review and the information the reviewer is asking was not visible because of DBPR. We write it like this: *All participants (i.e. survey respondents) gave*

their prior informed consent. The study has received ethical approval from the XXX Ethical Review Authority (permit XXX) and complied with GDPR.

Your lack of a rural-urban difference is an interesting find - although it reflects the fact that many rural areas are occupied by people of urban origin - such that the expected difference would be between "rurality" as a form of adherence to traditional activities and with a sense of place rather than rural living per se. Again, at least mention this issue.

A variety of recent studies related to wildlife have found little to no effect of rural residency or identity. For example, a recent study (Carlson, S. C., Dietsch, A. M., Slagle, K. M., & Bruskotter, J. T. (2023). Effect of semantics in the study of tolerance for wolves. *Conservation biology*, 37(2), e14003) that examined five different metrics 'tolerance' for wolves in the United States found no effect of residence in a Metropolitan Statistical Area in any of the five models. In contrast, that study found that identifying with interest groups often associated with rurality (e.g., hunter, farmer or rancher) or urbanity (e.g., animal rights) showed consistent effects across different measures of tolerance. This suggests that, mechanistically, it is not the location of one's residence but rather, one's interests, that determine attitudes toward (or tolerance of) wildlife. Likewise, another recent study (Vucetich, J. A., Bruskotter, J. T., Wilson, R., Elbroch, L. M., Feltz, A., & Offer-Westort, T. (2025). Support for the US Endangered Species Act Is High and Steady Over the Past Three Decades. *Conservation Letters*, 18(3), e13111) that examined support for the U.S. Endangered Species Act, which is often characterized as 'polarizing' and disproportionately adversely impacting rural communities, found no differences in support/opposition for the ESA across a rural-urban gradient. We have added this clarification ", **although our survey focused on rural residents and not on economic interests associated with "rurality" such as hunting and agriculture**" lines 95–96.

The "delisting drift" section is a bit biased. The point is worth mentioning in the wider context of the interest in following this first ever delisting of a species - but it could go either way - if it really helps then it would not be a bad thing for other species to follow - but if it all goes wrong then maybe wolves could be relisted.

We have added the example of Poland,: "**wolves in Poland have always been listed in Annex V but still experienced a strong population recovery.**" Lines 185–186.

It is also important to point out that your survey covers 3(4) species but the delisting is only wolves - and only changes things in a portion of countries because many others already had it on a lower protection status.

We clarify this lines 34–36: "*in June 2025, the wolf was moved from Annex IV (strict protection) to Annex V (permitting regulated exploitation) across all Europe (legal status did not change in the few countries where the wolf was already in Annex V).*" We also say line 194 "*it may set a precedent for managing other recovering species than the wolf*".

The majority vs interest group influence is interesting - but then it might also be useful to point out that when most people are indifferent to carnivore issues it would seem fair for those interest groups most engaged to have a larger say in policy? At least discuss.

This comment relates to who should have more weight in conservation politics: the general public, affected communities, or even future generations. This has been discussed at length in the literature and perspectives differ between authors and also between countries (i.e. between latin and anglo-saxon countries). Since we have limited space and this is not the focus of the study, we have only added a proposition and a reference: “ ***an issue at the core of contemporary conservation politics (Treves et al. 2017)***”. (line 149)

Finally, make the argument linking the study findings and support for EU policy more clear. None of your questions asked about it explicitly - so it is implicit in the interpretation of the questions - so tidy it up a little to be clearer.

We have tidied up. Since none of the questions asked about it explicitly, we have removed this argument from the title and the abstract. And we now say “*the downgrading the legal protection status of the wolf in the Habitats Directive [...] appears broadly aligned with public opinion across Europe if it prevents further population increases (and does not lead to decreases)*”. “ lines 173–176.

Reviewer #3:

In terms of content, some parts of the paper focus specifically on wolves, while others include bears and lynx as well, which creates a degree of confusion. I recommend either narrowing the focus exclusively to wolves or broadening the scope to consistently include all three large carnivores throughout the paper. If the latter approach is chosen, the title should be modified accordingly to reflect this broader focus.

We chose the latter and have changed the title to: “*Europeans support large carnivore recovery while opposing both further population growth and hunting*”

Regarding the categorization of respondents, I understand that individuals could self-identify as living in a large urban area, city, or village. However, it would be helpful for the authors to clarify how they have defined and categorized rural versus urban areas. For instance, I would consider a village to be more suburban than rural, depending on population density.

We did not define rural / urban areas on purpose because there was no straightforward classification that would apply consistently across Europe for this survey. For example, a ‘rural’ area as perceived by someone in the Netherlands, may be perceived as an urban area by a resident of Finland despite having similar human population densities.

Additionally, it is unclear whether quotas were used during sampling to ensure a representative distribution of respondents or if this was examined retrospectively.

Quotas were used during sampling to ensure a representative distribution of respondents, there were no retrospective post-hoc adjustments.

It is also worth noting, as established in previous research, that panel surveys often have a bias toward individuals with higher levels of education. This demographic skew may mean that the respondent pool is more representative of urban populations, and even if some individuals report living in rural areas, they may not accurately represent key rural groups such as shepherds, livestock owners, or farmers.

This comment related to 'ruralness' been understood as identification with interest groups associated with rural areas (see Carlson et al. 2023 for evidence) and is similar to a comment by reviewer 2 which we replied above. We have added this clarification “, ***although our survey focused on rural residents and not on economic interests associated with “rurality” such as hunting and agriculture***” lines 95–96.

I am also interested in how the age groups were categorized—what criteria or threshold were used to define "young" versus "old"?

We used the country specific median age reported by EuroStat. Supplementary table (Table S2) presents data on median age and % rural.

Finally, I would like to understand the rationale behind asking a different type of question about bears (i.e., the potential for killing humans), rather than maintaining a consistent focus on impacts such as damage to small domestic livestock (e.g., sheep, chickens, or beehives).

Our efforts were not aimed at making direct comparisons across species, but rather, capturing the types of events that are more likely to lead to change of attitudes. Because wolf attacks on humans are relatively rare, and evidence indicates wolves are less feared than bears (see the work of Maria Johansson and colleagues), we chose different conflict scenarios for these species.

In the conclusions, I would like to see more direct recommendations based on the findings, rather than focusing solely on the challenges that lie ahead.

We have added this new concluding paragraph: “***As the first-ever removal of strict protection of a species in Europe following a declared recovery, the downlisting of the wolf represents a policy experiment whose outcome remains uncertain. If it contributes to reduced conflicts and demonstrates that a more flexible legal framework can support coexistence, it may set a precedent for managing other recovering species. However, if it results in ecological backsliding or renewed controversy, reinstating a strict protection may become necessary—though such a reversal would not be automatic and would require political momentum contrary to that which enabled the downlisting.***” We also mention “***state-of-the-art monitoring of***

wolves across Europe and science-based management “ in the preceding paragraph (lines 186–187).

Responses to reviewers NATECOLEVOL-25061890A

Reviewer #1:

The abstract does not list any other carnivores than wolves. Given that you decided to stay with the title of carnivores after reviewer comments, I would suggest to also name European lynx and brown bear somewhere in the abstract.

We agree and now mention lynx and bears in the abstract.

P.3 line 88 - you mention 'compatriots' but I don't think this is actually correct. The panels do not typically distinguish between residents and nationals. Unless you specifically selected only nationals of a sampled country, please replace this with another term that does not indicate nationality.

We have corrected and replaced by 'respondents'.

Reviewer #3:

Rural–urban definition:

I understand that each country has its own criteria for defining what is urban and what is rural, which makes sense. However, I think I did not explain myself clearly earlier: How exactly did you divide the sample for the rural–urban analysis, based on the question included in the questionnaire?

Whether a respondent enters the category rural or urban is self-assessed by the respondent (question Q9, with answers “Village” or “Isolated house” converting to rural, and answers “Large urban area” or “City” converting to urban). We have added this clarification in the Supplementary Methods.

Panel issues and potential bias in urban–rural differences:

Again, I may not have been clear before. My comment was about possible panel bias, specifically that rural respondents who participate in panels often tend to have a higher level of education than the general rural population. Did you control for this factor? In other studies, researchers have found little to no difference between urban and rural responses due to this effect. I believe it would be important to add a limitation regarding the use of panels. For example, see Martínez-Jauregui et al. (2023):

<https://doi.org/10.1016/j.jenvman.2023.117236>

We did not control for this factor and acknowledged it in the Supplementary Methods, section Panelist recruitment. We have now made this more explicit and cite a recent paper on the limits of non-probability samples:

The common limitations of non-probability samples (i.e. not all individual in the targeted population have a known and non-zero probability to be sampled) apply to our study (Freese, J. & Jin, O. 2025. Online Nonprobability Samples. *Annual Review of Sociology* 51, 109–128). It is possible, for example, that individuals with a higher level of education may be overrepresented in our panels versus the population, especially in rural areas.

The reviewer points us toward a paper (Martínez-Jauregui, M., Delibes-Mateos, M., Arroyo, B., Glikman, J. A., & Soliño, M. (2023). Beyond rural vs urban differences: A close match in european preferences in some basic wildlife management and conservation principles. *Journal of Environmental Management*, 331, 117236.) but the word “educa” (as in educated, education, educate) does not appear anywhere in the publication. We therefore do not cite it.

Age definition:

What is the threshold for being considered “young”? I still do not understand how the authors divided the sample into “young” and “old” groups based on the categorical question.

We used the median age for each country, shown in Table S2 and extracted from Eurostat: Median age by sex in 2021. https://doi.org/10.2908/EQ_POP04. Young is having an age smaller than the median age, old is having an age larger than the median age. Age is obtained from birth year documented in question Q10.

Editor

please ensure that "large-carnivore" is hyphenated consistently through the text

Although many style guidelines would recommend hyphenating “large-carnivore” when used as an adjectival phrase before a noun, in practice I have generally not seen this in the literature. I have conducted a quick quantitative assessment from the full text of three books on the topic:

Clark, S.G., Rutherford, M.B., 2014. *Large Carnivore Conservation: Integrating Science and Policy in the North American West*. University of Chicago Press. “large carnivore” occurs **238** times, “large-carnivore” occurs **0** times.

Clark, T.W., Rutherford, M.B., Casey, D. (Eds.), 2010. *Coexisting with large carnivores: lessons from Greater Yellowstone*. Island Press, Washington. “large carnivore” occurs **389** times, “large-carnivore” occurs **0** times.

Hovardas, T., Taylor and Francis (Eds.), 2018. *Large carnivore conservation and management: human dimensions*, First edition. ed, Routledge library editions. welfare and the state. Routledge. “large carnivore” occurs **593** times, “large-carnivore” occurs **2** times (in URLs).

Ray, J.C. (Ed.), 2005. Large carnivores and the conservation of biodiversity. Island Press, Washington. "large carnivore" occurs **513** times, "large-carnivore" occurs **1** time (in a table).

I have therefore *not* hyphenated "large carnivore". The expression "large carnivore" is well accepted and I think it is clear from context in our article that "large carnivore conservation" should not be understood as "large conservation of carnivores".